# Dynamic Response of a Single-Pier Jacking of a Continuous Box Girder Bridge Based on Vehicle–Bridge Coupling

**Ziyang Ye \*, Changmin Yang and Honggang Li**

College of Civil Engineering and Architecture, Hebei University, Baoding 071000, China;
changmyang@126.com (C.Y.); lhg199715@163.com (H.L.)
* Correspondence: 13309644096@163.com

**Abstract:** In order to improve the bearing capacity of a bridge under dynamic impact and at the same time to improve the durability and safety of the bridge, the bridge should be jacked up, and most of the current research on bridge jacking upgrading is on synchronous jacking upgrading. There has been little research on single-pier jacking upgrading, especially the dynamics response problem under single-pier jacking upgrading of the bridge. This paper is based on the principle of vehicle–bridge coupling, and the vehicle was analyzed at different speeds by using the established model, vehicle weight, and multi-lane driving, comparing the dynamic response of the bridge before and after the single-pier jacking retrofit. The study analyzed the impact of the bridge single-pier jacking retrofit on the bridge impact coefficient, and evaluated the impact coefficient of the bridge from the perspective of dynamics. A comprehensive evaluation of the bridge single-pier jacking retrofit project was carried out. The following conclusions are drawn: the single-pier jacking modification of the bridge increased the bridge's capacity under dynamic impact and also enhanced the durability and safety of the bridge.

**Keywords:** vehicle–bridge coupling; continuous box girder bridge; single-pier jacking; impact factor





## 1. Introduction

At present, most of the bridge jacking retrofit projects in China focus on the synchronous jacking of bridges, while less attention is paid to non-synchronous jacking; the single-pier jacking of a bridge produces a significant displacement difference in the direction of the bridge's course. The bridge has thereby been modified by the single-pier jacking, the bridge's course line has changed, and this may produce a significant effect when transport vehicles travels on the bridge. In this context, this paper considers the coupling of cars and bridges. Therefore, this paper investigates the dynamic response of single-pier jacking of a prestressed continuous girder bridge against the background of single-pier jacking of the bridge under the coupling effect of vehicle and bridge.

In the field of bridge jacking problem research, Du Baisong, Luo Birong et al. [1] used finite element software to analyze the effects of different bridge jacking positions and other bridge jacking methods on bridge stresses. Niu Jianguang et al. [2] selected an overall jacking scheme for a ramp-bearing replacement project by comparing different jacking schemes. China Communications Construction Group [3], aiming at the operation characteristics of high-speed railways without fractured bridges, developed large tonnage bearing replacement equipment for replacing the bridge bearings of Lou Tunnel Bridge after jacking the bridge to a certain height. Yan Xingfei et al. [4] used finite element software to simulate the force state of the bearing in the pier body after it was cut when the main jack and the subsequent jack worked alternately. Wu Yaodong et al. [5] simplified bridge analysis using Midas software to analyse the bridge forces under different jacking schemes by changing the jacking sequence, single jacking height, and the number of jacking times. Liu Jashun [6] studied the effect of synchronous jacking on the bridge and compared it with a displacement synchronous jacking scheme, and it was found that the effect of synchronous jacking on the bridge was greater than that of displacement synchronous jacking.

Liu Jianwei, Li Dejian et al. [7] investigated the effect of transverse displacement difference on the bridge force state during bridge jacking construction. Li Feng et al. [8] calculated the conditions of asynchronous bridge jacking using 3D software. They investigated the relationship between the vertical displacement difference of asynchronous bridge jacking and the deformation behavior of the girders. Hu Danyi [9], based on a rectangular shield bridge jacking project in an ultra-shallow soft soil layer, reported that the fundamental law of pavement settlement over time was summarized by the monitoring data of the on-site test section.

Bridge jacking construction has a significant impact on the bridge's linear shape. When transport vehicles travelling on the bridge may have a significant impact, it is necessary to analyze the vehicle–bridge coupling dynamics in the latter's operational process. Shen Ruli et al. [10] analyzed the dynamic response of an axle–bus coupling system when a high-speed passenger car passed through a supported beam bridge and studied the relationship between the resonance-producing vehicle speed and the bridge characteristics. Shan Deshan et al. [11] investigated the numerical simulation method of vehicle–bridge coupling and implemented the algorithmic program by computer. Lin Yusen et al. [12] studied the dynamic response of trains to excitation generated by track unevenness based on vehicle–bridge coupling theory. Cheng Baorong et al. [13] reported that the vehicle–bridge coupling equations were solved by reducing the system degrees of freedom using the modal synthesis technique. Li Wusheng et al. [14] considered the car and bridge as two subsystems and realized the coupling relationship between the two subsystems. Chen Zhaowei et al. [15] studied the elastic wheel–metro train–LSCSB coupling system in depth. Lu Zhourui et al. [16] analyzed the coupling problem between maglev vehicles and bridges in the case of track unevenness. Han Wanshui et al. [17] gave the critical factors for developing coupled vibration systems for windmill bridges. Gara F et al. [18] investigated the effect on the modal parameters of bridges under the dynamic impact of trucks. Nishimura et al. [19,20] carried out theoretical analysis and experimental studies on the derailment mechanism and operational safety of high-speed railway vehicles under seismic and track unevenness conditions.

There are several studies in the literature on the dynamic interactions between moveable vehicles and bridges, mainly devoted to investigating the changes in the dynamic behavior of bridges induced by vehicles travelling on the road, depending on the road and vehicle characteristics, such as road unevenness and vehicle speed [21–24]. Recently, researchers have also investigated how vehicle–bridge interactions affect the dynamic response of bridges during earthquakes [25,26].

Although there has been much research on the dynamic interaction between vehicles and bridges, most of the research on the bridge jacking problem relates to the synchronous jacking problem of bridges. Studies on the single-pier jacking of bridges are few and far between. Even when they are present, the static performance of bridges during the jacking of a single pier has been studied and analyzed, but there are not many discussions on the dynamic performance of bridges before and after the deformation of the jacking. In this paper, the bridge is compared and analyzed before and after single-pier jacking and according to the principle of vehicle–bridge coupling. The dynamic response of the bridge is analyzed with vehicles travelling at different speeds, weights, and in multiple lanes by using the established model. The dynamic response of the bridge before and after the jacking transformation of the single pier is compared, and finally, the effect of the jacking transformation of the bridge on the impact coefficient of the bridge is analyzed, assessing the dynamic performance of the bridge from the kinetic point of view.

## 2. Establishment of Coupled Vehicle-Bridge Vibration Equations

### 2.1. Vehicle Modelling

Brazil and other countries have proposed an international smoothness index, IRI, by using a two-degree-of-freedom planar vehicle model for road surface unevenness tests and calculating the index from the vertical displacement of the vehicle as the two-degree-of-

freedom planar vehicle model passes over the roadway [27,28]. As the vehicles travelling on the bridge in this paper are mainly integral four-axle transporters, this paper takes a four-axle, eleven-degree-of-freedom model [29]. The vehicle consists of eight wheels and a body with a total of nine rigid bodies. The four-axis space vehicle model includes the pitch motion of the vehicle body vibrating in the vertical direction, the roll motion of the vehicle body rotating in the longitudinal direction, the yaw of the vehicle body rotating in the transverse direction, and the motion of the eight wheels in the vertical direction of the vehicle body.

As shown in Figure 1, $a$, $b$, $c$, $d$ is the distance from each axle to the centre of mass of the vehicle body, $I_{fy}$ is the vehicle body pitch moment of inertia, $I_{cg}$ is the moment of inertia of the side roll of the vehicle, $M_1 \sim M_8$ is the weight of the wheels, $M_9 \sim M_{16}$ is the weight of each axle of the vehicle, $M_c$ is the mass of the vehicle body.

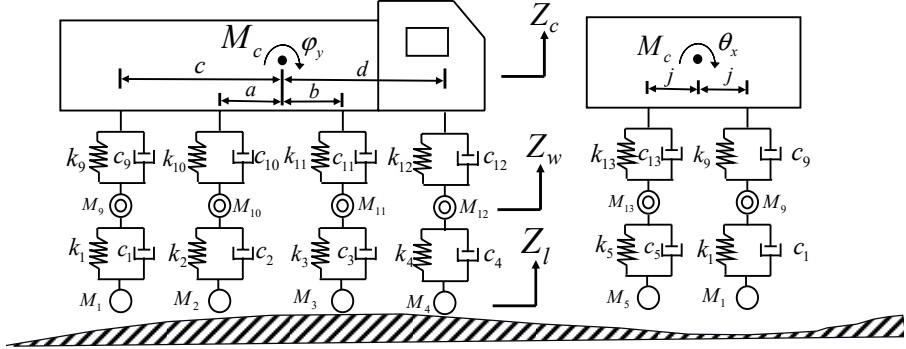

**Figure 1.** Four-axis space vehicle model of degrees of freedom.

The vehicle system dynamics equations can be developed according to the D'Alembert principle:

$$[M_v^t]\left\{\ddot{Z}_v^t\right\} + [C_v^t]\left\{\dot{Z}_v^t\right\} + [K_v^t]\left\{Z_v^t\right\} = \left\{F_v^t\right\} \tag{1}$$

In Equation (1), $[F_v^t]$ is the excitation force to which the vehicle is subjected, $[Z_v^t]$ the displacement vector of the vehicle components, $[M_v^t]$, $[C_v^t]$, $[K_v^t]$ the mass, damping and stiffness matrix of the vehicle.

### 2.2. Bridge Vibration Equations

According to D'Alembert's principle, the structural vibration differential equations for the bridge can be established, which are organized into matrix form as:

$$[M_b]\left\{\ddot{Z}_b^t\right\} + [C_b^t]\left\{\dot{Z}_b^t\right\} + [K_b]\left\{Z_b^t\right\} = \{F_b\} \tag{2}$$

In Equation (2), $\left\{Z_b^t\right\}$ is the displacement vector of the bridge, $[M_b]$ is the mass matrix of the bridge, $[C_b]$ is the damping matrix of the bridge, $[K_b]$ is the bridge stiffness matrix, $\{F_b\}$ is the overall external load vector of the bridge.

### 2.3. Road Surface Unevenness Model

Bridge deck irregularities can be viewed as a normal stochastic process with anisotropic ephemerality [30]. The international description of its irregular elevation is usually expressed using the frequency domain method and power spectral density function. This is shown in the following equation:

$$G_q(n) = G_q(n_0)\left|\frac{n}{n_0}\right|^{-w} \tag{3}$$

In Equation (3), $n_0 = 0.1$ m$^{-1}$ is the spatial reference frequency, $G_q(n_0)$ is the power spectral density of a particular unevenness class of pavement, $w$ is the frequency index which is generally taken as $w = 2$, $n$ denotes a particular frequency among the valid frequencies.

According to the national regulations on pavement irregularity [20], pavement unevenness can be divided into eight levels from A to H. This paper's study on pavement unevenness only considers the three levels from A to C.

The unevenness of the bridge deck can be regarded as a random function. This function is subject to a normal distribution, so in the generation of different levels of unevenness, data can be used to generate the unevenness of the bridge deck samples. According to the power spectrum density function of each level of the unevenness of the bridge deck, its unevenness samples can be expressed as:

$$r(x) = \sum_{j=1}^{m} \sqrt{2G(n_j)} sin(2\pi x n_j + \theta_j) \tag{4}$$

In Equation (4), $x$ is the displacement along the cis-bridge direction, $m$ is the number of segments divided by spatial frequency, $G(n_j)$ is the $j$ frequency spectral density corresponding to the median spatial frequency of the segment, $n_j$ is the median spatial frequency of segment $j$, and $j$ is a uniformly distributed random variable over $[0, 2\pi]$.

As the level of unevenness increases, the undulation of the road surface becomes greater. When performing a vehicle–bridge coupling analysis, the degree of unevenness of the bridge deck significantly influences the excitation caused by the vehicle on the bridge deck. Hence, the degree of unevenness of the bridge deck is an influencing factor that cannot be ignored in a vehicle–bridge coupling analysis.

### 2.4. Implementation of the Vehicle–Axle Coupling Model

According to Newton's second law, the vehicle interacts with the bridge, and the forces on the contact surface between the wheels and the bridge are of the same magnitude and opposite directions. The excitation forces on the bridge deck include the gravitational force of the vehicle and the force of the suspension system acting on the bridge deck. So, the vertical displacement of the vehicle's wheels and the vertical displacement of the bridge satisfy the relationship Equation (5):

$$Z_{li} = Z_{bi} + Z_{pi} + Z_{\omega} \tag{5}$$

In Equation (5), $Z_{li}$ is the vertical displacement of wheel $i$, $Z_{bi}$ is the displacement of the bridge floor where the wheel $i$ is located, $Z_{pi}$ is the irregularity displacement of the bridge floor where the wheel $i$ is located, and $Z_{\omega}$ is the deformation of the bridge where the wheel $i$ is located.

### 3. Bridge Vibration Response Based on Vehicle–Bridge Coupling

The single-pier jacking of bridges can cause large deformations, so the single-pier jacking of bridges is a severe test for bridges. Previous jacking cases have studied and analyzed the static performance of bridges only during single-pier jacking, but the operation of bridges after jacking modifications has not been discussed much. This chapter analyses the vibration response of the bridge before and after single-pier jacking and, based on the vehicle–bridge coupling principle, uses the constructed model to analyze the dynamic response of the bridge when the vehicle is travelling at different speeds, vehicle weights, lanes, and in multi-lane traffic. It furthermore compares the dynamic response of the bridge before and after single-pier jacking and provides a comprehensive evaluation of the bridge single-pier jacking retrofit project from the perspective of the dynamics.

### 3.1. Finite Element Modelling

#### 3.1.1. Vehicle Modelling

The four-axle vehicle used in this paper is the China National Heavy-Duty Truck HOWO-T6G four-axle dump truck; the vehicle model can be simulated in Ansys APDL using a moving mass model, with the body mass expressed in Mass21 units and the suspension system expressed in Combin14 with damping parameters and elastic mode parameters. The standard methods used for coupling the vehicle to the bridge in the model are the displacement coupling method, the birth–death cell method and the displacement contact method. The displacement coupling method is used more frequently, and in this paper, the displacement coupling method is used to realize the coupling between the vehicle and the bridge; after the finite element meshing is completed, contact pairs are established between the wheels and the bridge. The vehicle model in Ansys APDL is shown in Figure 2.

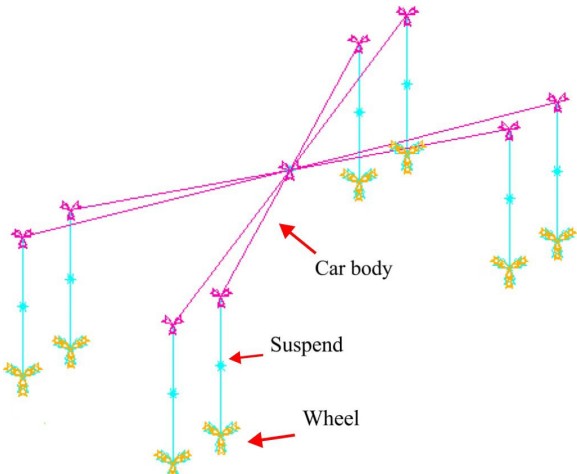

**Figure 2.** Ansys four-axis vehicle model.

#### 3.1.2. Bridge Modelling

The bridge has four spans and a total length of 160 m (40 m + 40 m + 40 m + 40 m) with a box girder width of 30 m and a height of 2.2 m. It is a single-box multi-chamber pre-stressed cast-in-place continuous box girder bridge. The bridge was cast in C50 concrete, and the cover girders in C30 concrete. The box girder section of the bridge is shown in Figure 3. The elevation is shown in Figure 4. Z0#~Z4# are the abutments and piers of the bridge.

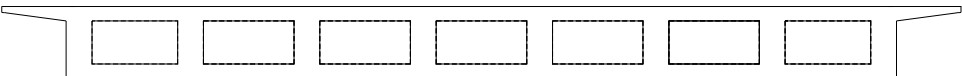

**Figure 3.** Box girder cross-section.

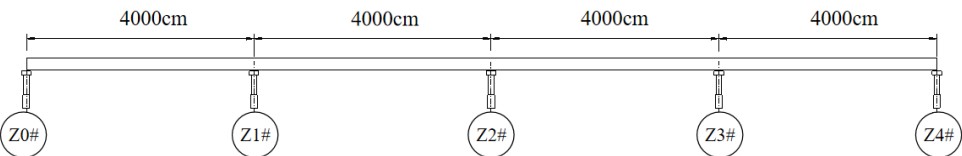

**Figure 4.** Bridge elevations.

The Solid65 unit has 24 degrees of freedom, i.e., each of the eight nodes of the Solid65 unit has 3 degrees of freedom in the x, y and z directions. Solid65 units can simulate problems such as the cracking of concrete. According to the research content of this paper, the Solid65 cell was chosen to build the finite element model of the bridge.

Ansys was used to build the solid model of the bridge. The Solid65 unit with eight nodes and twenty-four degrees of freedom was selected for the simulation. The bridge has 25 bearings, including 16 bi-directional bearings, 4 movable bearings in the direction of the bridge, and one fixed bearing. This study applied a forced displacement load at the bearings to simulate the jacking of a single bridge pier. The bridge model is shown in Figure 5:

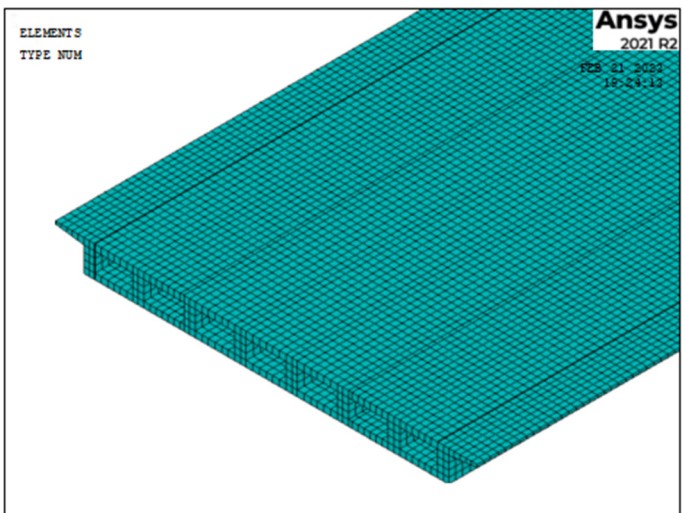

**Figure 5.** Finite element model of box girder.

In the process of bridge pier abutment, the bridge undergoes different degrees of settlement. The meaning of "degree of settlement" is the deformation caused by the internal and external factors of the bridge; through the detection of Z0#, Z4#, two abutments settlement amounted to 101 mm, 120 mm, respectively, the abutment of the settlement in the range of 1–4 cm. Thus, it is possible to obtain a significant difference in height between the bridge and the road connection, which makes it impassable for vehicles. Vehicles cannot pass. Only the bridge's Z0# and Z4# abutments were jacked up in the bridge reconstruction project. The bridge's specific settlement data are shown in Table 1, below.

**Table 1.** Settlement values for the bridge.

| Pier Number | Z0# | | | Z1# | | | Z2# | | | Z3# | | | Z4# | | |
|---|---|---|---|---|---|---|---|---|---|---|---|---|---|---|---|
| Bridge pier location | 1 | 3 | 5 | 1 | 3 | 5 | 1 | 3 | 5 | 1 | 3 | 5 | 1 | 3 | 5 |
| Settlement/(mm) | 107 | 101 | 95 | 37 | 34 | 25 | 34 | 22 | 18 | 37 | 30 | 22 | 124 | 121 | 116 |
| Average settlement/(mm) | | 101 | | | 32 | | | 25 | | | 30 | | | 120 | |
| Relative Settlement/(mm) | | 69 | | | 7 | | | -- | | | 5 | | | 91 | |

### 3.2. Bridge Vibration Response Based on Vehicle–Bridge Coupling

From the data, it can be seen that the bridge Z4# abutment single-pier jacking transformation had the most significant impact on the fourth span. Therefore, in this section, only the acceleration, velocity, and dynamic deflection of the fourth span mid-span node and 1/4 node are analyzed for the vehicle–bridge coupling. According to China's regulations on pavement unevenness [31], pavement unevenness is divided into eight grades from A to H. Normal pavement is usually regarded as unevenness grade A. In the following sections, if the level of unevenness, speed, or load is not specified, the default level of unevenness is A, the speed is 60 km/h, and the load is complete; as the vehicles driving on this bridge are mostly four-axle transport vehicles, this study takes four-axle transport vehicles as the research object.

### 3.2.1. Bridge Vibration Response under the Action of Traffic at Different Speeds

In the vehicle–bridge coupling system, vehicles are one of the primary sources of bridge excitation, of which vehicle travel speed is a non-negligible element. In order to analyze the influence of a four-axle transport vehicle on a bridge at different speeds of the transport vehicle when fully loaded, this study set up four working conditions of a fully loaded transport vehicle travelling at 30 km/h, 40 km/h, 50 km/h, and 60 km/h in the middle lane, respectively. The dynamic deflections of the bridge span nodes at different vehicle speeds are shown in Figures 6–9.

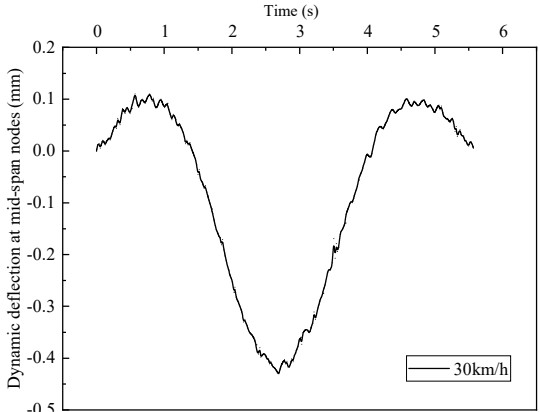

**Figure 6.** Dynamic deflection diagram of bridge cross-node at 30 km/h.

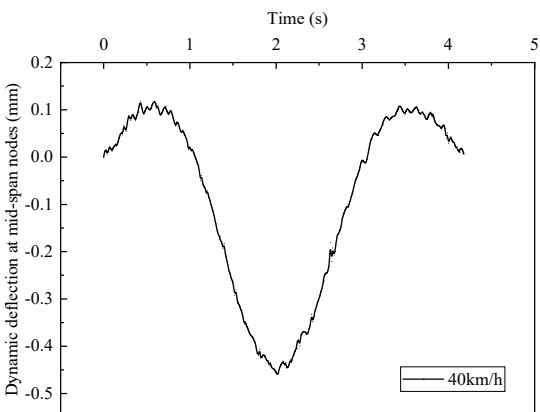

**Figure 7.** Dynamic deflection diagram of bridge cross-node at 40 km/h.

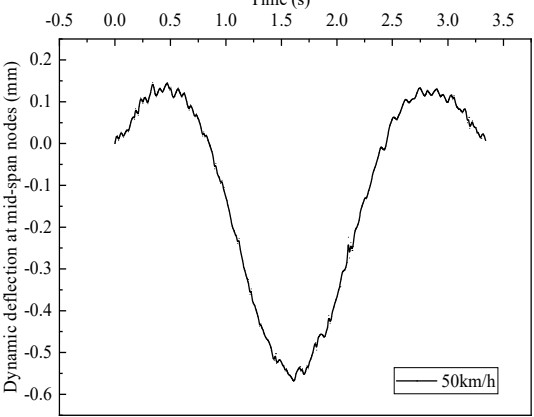

**Figure 8.** Dynamic deflection diagram of bridge cross-node at 50 km/h.

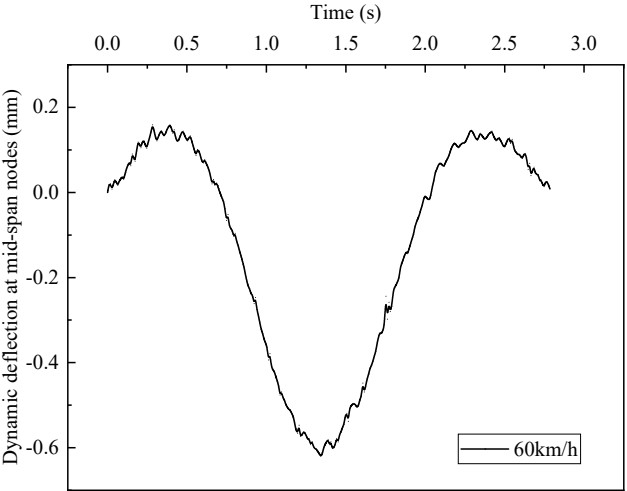

**Figure 9.** Dynamic deflection diagram of bridge cross-node at 60 km/h.

The peak dynamic deflections of the bridge and parapet span nodes at vehicle speeds of 30 km/h, 40 km/h, 50 km/h, and 60 km/h are collated in Table 2.

**Table 2.** Peak dynamic deflection at mid-span nodes of bridges at various speeds.

| Speed/(km/h) | 30 | 40 | 50 | 60 |
|---|---|---|---|---|
| Peak dynamic deflection at mid-span nodes/(mm) | 0.43 | 0.46 | 0.53 | 0.62 |
| Peak dynamic deflection in the guardrail span/(mm) | 0.35 | 0.37 | 0.41 | 0.48 |

As can be seen from Table 2, when the vehicle speed reaches 60 km/h, the dynamic deflection of the span node reaches 0.62 mm. In the case of constant vehicle load, as the speed increases, the dynamic deflection of the span node and the span node of the bridge guardrail also gradually increase, and the dynamic deflection of the guardrail node is smaller than the dynamic deflection of the span node. From vehicle speed 30 km/h to 60 km/h, span node dynamic deflection increased by 44.2%, guardrail span dynamic deflection increased by 37.1%, and the vehicle driving has little influence on the dynamic deflection of the guardrail span node.

The vertical velocities of the bridge span nodes when travelling at 30 km/h, 40 km/h, 50 km/h, and 60 km/h are shown in Figures 10–13:

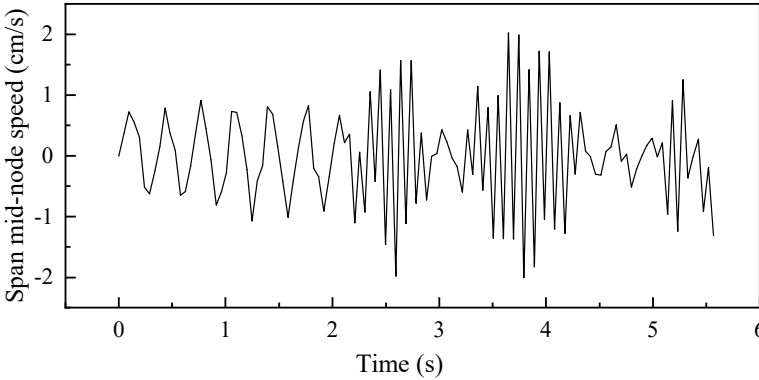

**Figure 10.** Vehicle speed 30 km/h.

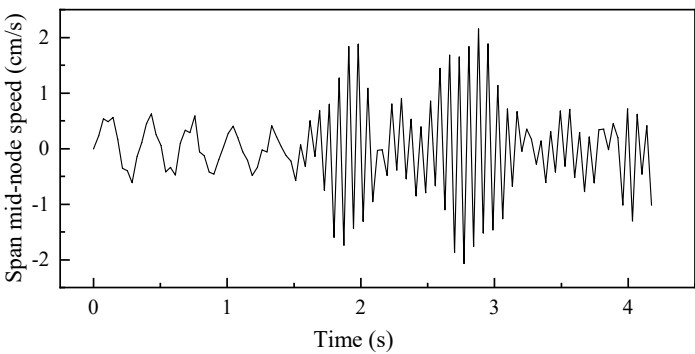

**Figure 11.** Vehicle speed 40 km/h.

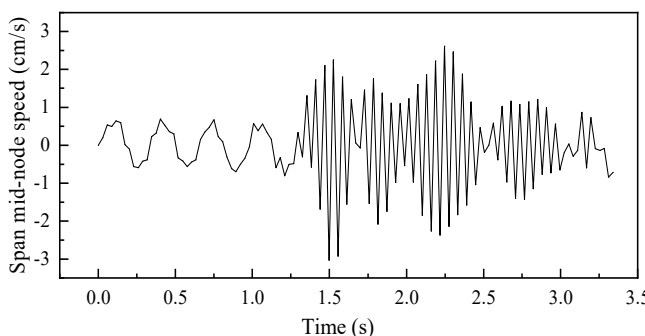

**Figure 12.** Vehicle speed 50 km/h.

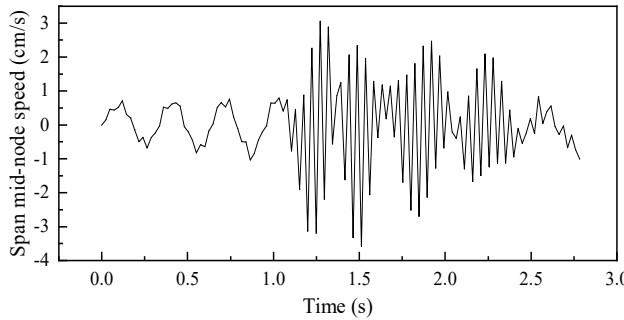

**Figure 13.** Vehicle speed 60 km/h.

The peak vertical velocities at the mid-span nodes of the bridge at different vehicle speeds are shown in Figure 14.

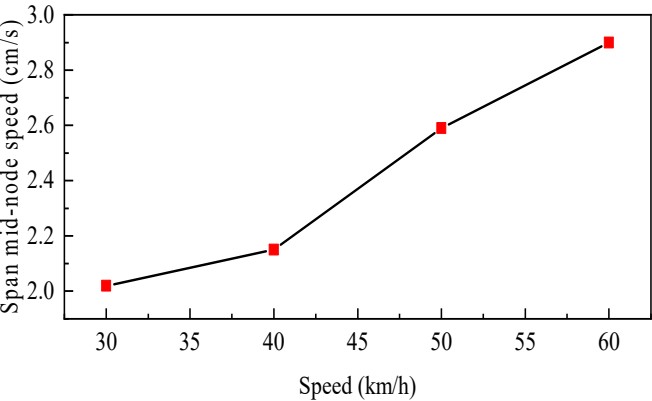

**Figure 14.** Peak velocity in the span of the bridge at different speeds.

As can be seen from Figure 14, the greater the vehicle speed, the greater the vertical velocity at the node in the bridge span. However, as the vehicle speed increases from 40 km/h to 50 km/h, it is clear that the vertical speed at the node in the span of the bridge changes more rapidly. When the vehicle speed is reduced from 30 km/h to 40 km/h, the speed change at the bridge span node is less.

### 3.2.2. Bridge Vibration Response under Different Vehicle Weights

The loads of four-axle transport vehicles are very high, and the overloading of actual engineering transport vehicles is frequent. In order to study the effect of a four-axle truck with different loadings on a bridge after a single-pier jacking modification, a four-axle truck was set up to drive in the middle lane at a maximum speed of 60 km/h under three conditions: empty, fully loaded, and overloaded by 15%, to analyze the effect of different loads of four-axle trucks on the bridge after a single-pier jacking.

The dynamic deflection of the nodes in the span of the bridge under different loads of vehicles is shown in Figure 15.

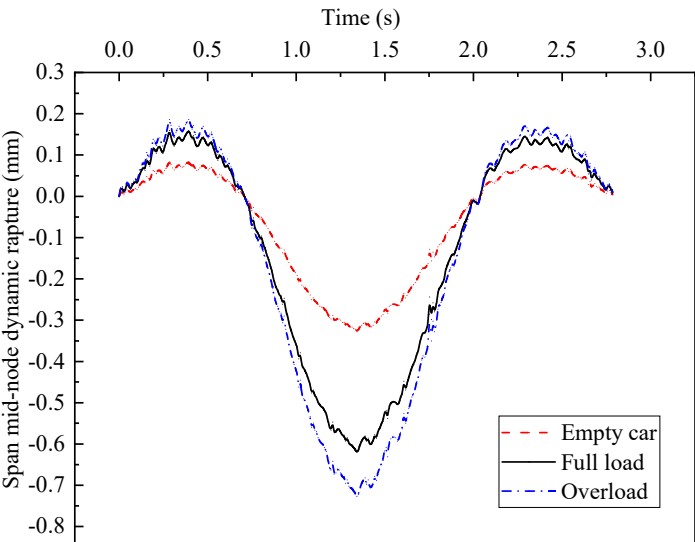

**Figure 15.** Dynamic deflections of bridge mid-span nodes for different vehicle weights.

The peak values of dynamic deflection for the periods when the vehicle is empty, fully loaded, and overloaded by 15% are 0.31 mm, 0.62 mm, and 0.72 mm, respectively, and it can be seen from the figure that the dynamic deflection of the bridge span nodes increases as the vehicle load increases. The variation of acceleration at the span nodes of the bridge under different vehicle loads is shown in Figure 16.

The peak accelerations at the span nodes of the bridge under the three operating conditions of empty, fully loaded, and overloaded by 15% are 2.58 m/s$^2$, 2.59 m/s$^2$ and 2.61 m/s$^2$, respectively. It can be seen from the graph that the acceleration at the span nodes of the bridge increases with the increase of the traffic weight, but the change is not very big.

### 3.2.3. Bridge Vibration Response under Multi-Lane Action

In the actual operation of the bridge, there are generally multiple lanes with vehicles in motion. In order to study the dynamic response of the bridge under multiple lanes of traffic, the impact of three working conditions on the bridge was calculated after the bridge was modified by single-pier jacking, with one vehicle in lane 6, two vehicles in lanes 5 and 6, respectively, and three vehicles in lanes 4, 5, and 6, respectively, with the vehicle arrangement for each working condition as shown in Figures 17–19.

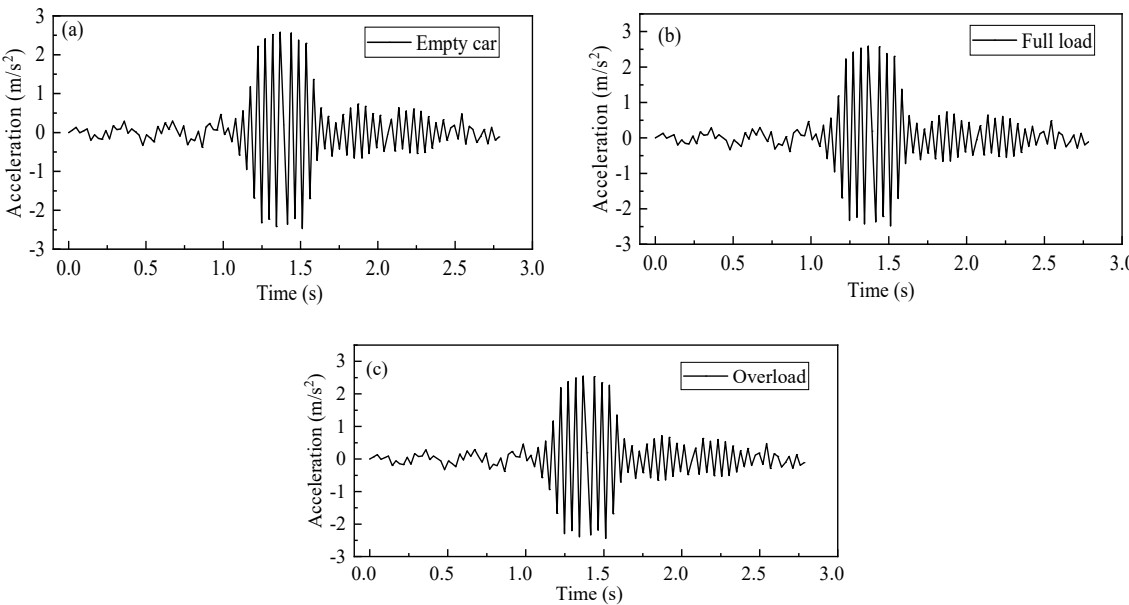

**Figure 16.** Dynamic deflection in the span of a bridge after jacking of a single pier with different vehicle weights. (**a**) Empty car. (**b**) Full load. (**c**) Overload.

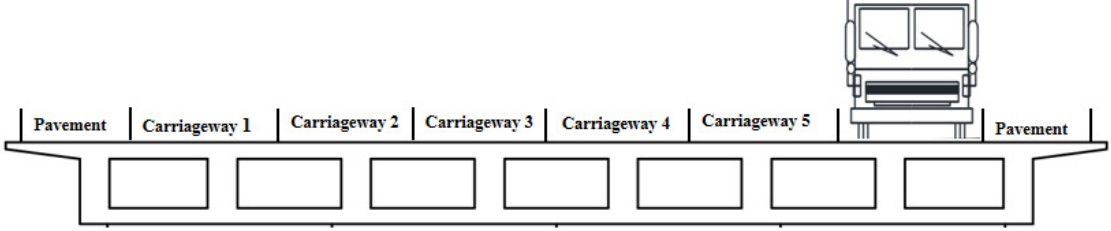

**Figure 17.** A vehicle travelling in lane 6.

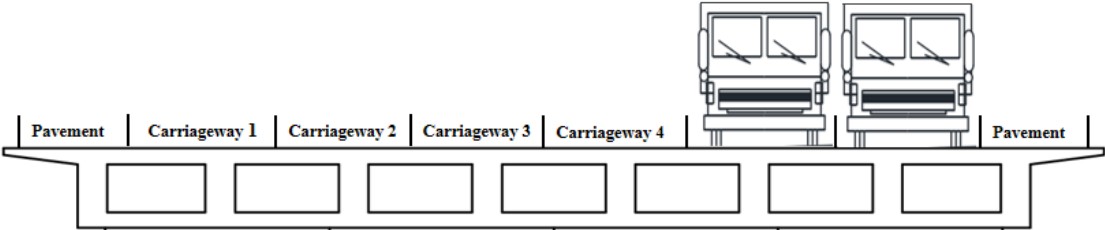

**Figure 18.** Two vehicles travelling in lanes 5 and 6.

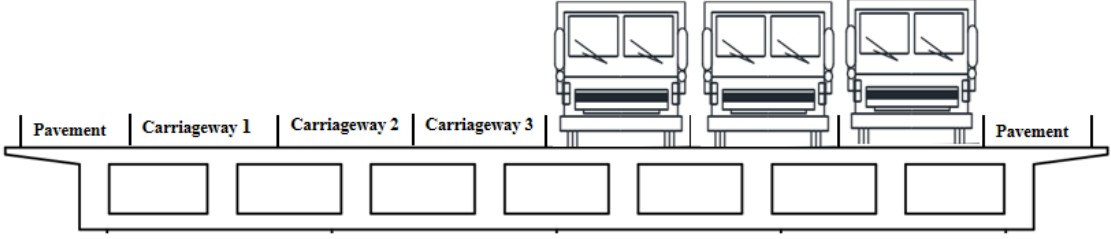

**Figure 19.** Three vehicles travelling in lanes 4, 5 and 6.

When driving vehicles in multiple lanes, it is assumed that all vehicles travel at a uniform speed and that travel speed remains consistent between vehicles. The peak dynamic deflections in the bridge span and guardrail span for different numbers of transverse vehicles are shown in Table 3.

**Table 3.** Peak dynamic deflection in bridge spans and parapet spans.

| Number | One Car | Two Cars | Three Vehicles |
|---|---|---|---|
| Spanwise dynamic deflection/(mm) | 0.54 | 0.59 | 0.72 |
| Guardrail side span dynamic deflection/(mm) | 0.66 | 0.75 | 0.81 |

The increase in the number of transverse vehicles leads to an increase in the dynamic deflection of the bridge; the dynamic deflection of the guardrail side span is greater than the dynamic deflection of the bridge's central axis span because the vehicle load is distributed from the outermost lane row to the inner side. The peak deflection values of the guardrail side span deflection are shown in Figure 20, as the number of vehicles travelling across the bridge gradually increases from one vehicle to three vehicles:

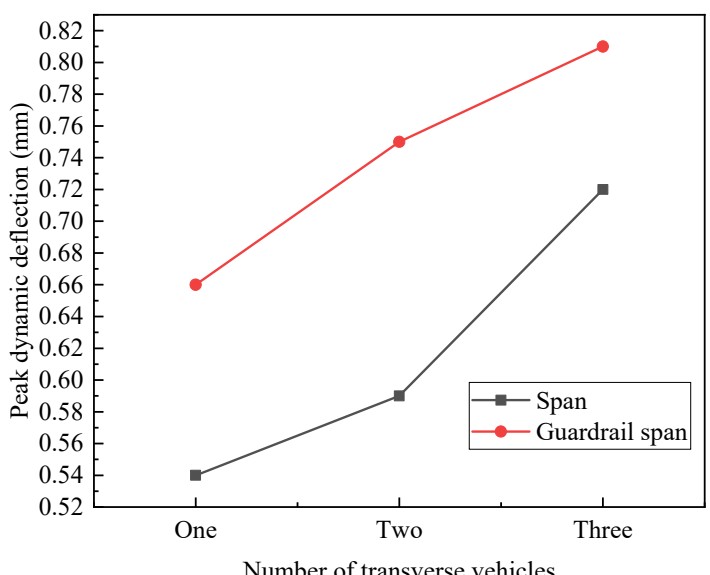

**Figure 20.** Variation of peak dynamic deflection in the span of bridge and parapet for different numbers of transverse vehicles.

As seen in Figure 20, as the number of vehicles travelling laterally increases, the increase in dynamic deflection at the guardrail side is less than the increase in dynamic deflection in the span. This is because the initial single vehicle is loaded onto the outermost lane 6. As the number increases, the loaded vehicles move closer to the span, resulting in a greater rate of increase in dynamic deflection.

### 3.2.4. Bridge Vibration Response at Different Levels of Smoothness

Among the factors influencing the vehicle–bridge coupling system, the unevenness of the bridge deck is also essential, in addition to the vibration characteristics of the vehicle and the bridge itself. In a coupled vehicle–bridge system, the excitation of the bridge deck is mainly caused by the vehicle and the random excitation is caused by the unevenness of the bridge deck when the vehicle is in motion, without taking into account the effects of external factors such as seismic and wind effects. The excitation caused by the unevenness of the bridge deck is random and is a significant influence in the coupled vehicle–bridge system. This section calculates the bridge vibration force response for bridge decks when smooth and with Class A, Class B, and Class C unevenness.

The dynamic deflections of the span nodes of the travelling bridge at different levels of unevenness of the bridge deck are shown in Figures 21–24.

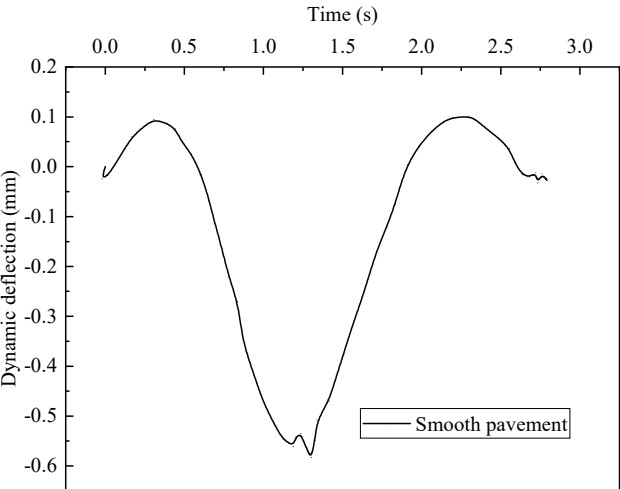

**Figure 21.** Smooth deck.

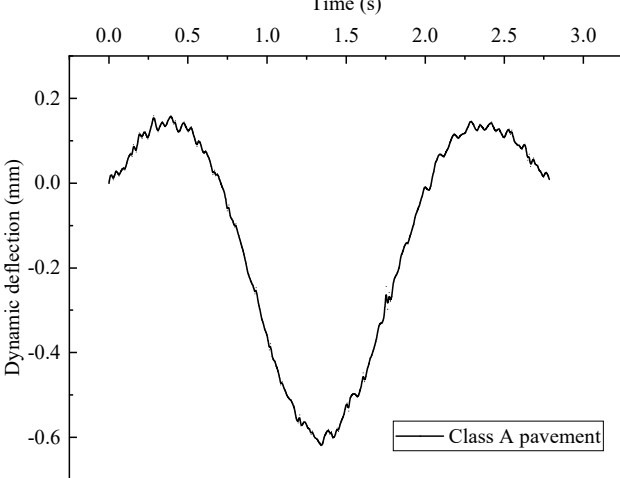

**Figure 22.** Class A deck.

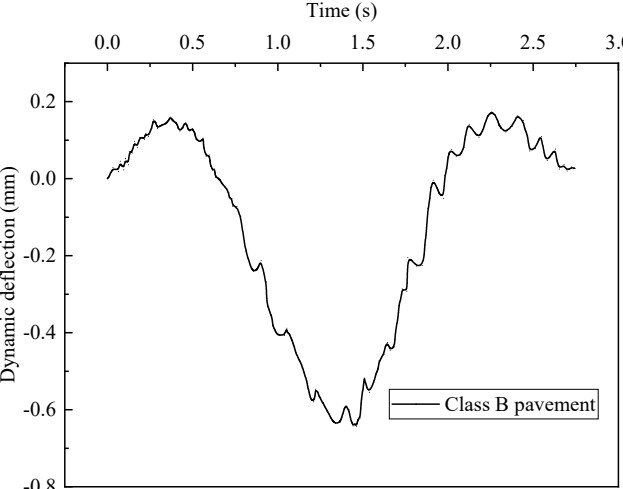

**Figure 23.** Class B deck.

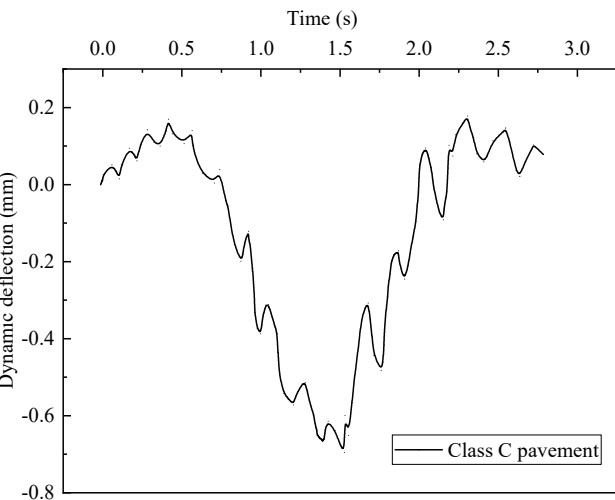

**Figure 24.** Class C deck.

When travelling on a bridge at a speed of 60 km/h, the poorer the smoothness of the bridge deck, the greater the excitation caused to the bridge by the vehicle movement. As can be seen from the graph above, as the smoothness of the bridge deck decreases, the more significant the change in dynamic deflection of the nodes in the bridge span. The peak dynamic deflection is 0.58 mm when the bridge deck is smooth and 0.62 mm, 0.64 mm, and 0.69 mm when the bridge deck is not smooth for classes A, B, and C, respectively.

### 3.2.5. Comparison of Bridge Dynamic Response before and after Single-Pier Jacking Modification

In order to determine the effect of the single-pier jacking modification on the bridge dynamic response, the bridge dynamic response to a vehicle travelling before the bridge jacking modification was simulated using Ansys APDL and compared with the results. The operating conditions set in this section were a fully loaded transport vehicle travelling on the bridge at 60 km/h. The results of the comparison are shown in Figure 25:

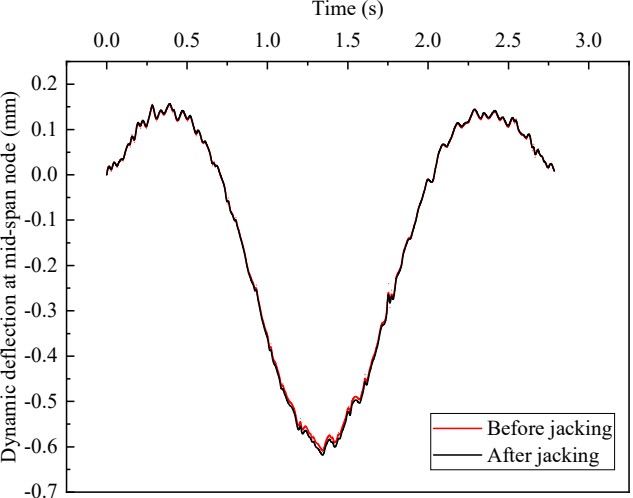

**Figure 25.** Dynamic deflection at the mid-span node before and after jacking of a single pier of the bridge.

As shown in Figure 25, after the single-pier jacking modification, the dynamic deflection of the bridge increased from 0.6105 mm before jacking to 0.62 mm after jacking, but the increase was only 1.56%. Therefore, the single-pier jacking of the bridge had no significant

effect on the excitation of the bridge when the vehicle was travelling on the bridge and that effect could be ignored.

## 4. Effect of Bridge Jacking Modifications on Impact Factors

In order to calculate the effect of single-pier jacking modification on the impact coefficient of the bridge, the impact coefficient of the bridge before and after the single-pier jacking modification was calculated according to the definition of impact coefficient in the specification and the data obtained through finite element model simulation.

According to the definition of the impact factor in the code, $\mu$ is the impact factor and $(1 + \mu)$ is the coefficient of increase of the dynamic effect:

$$\mu = \left| \frac{U_d}{U_j} \right| - 1 \tag{6}$$

In Equation (6), $U_d$ is peak deflection under dynamic load, $U_J$ is peak bridge deflection under static load.

$U_d$ takes the maximum dynamic deflection on the bridge dynamic deflection diagram when the vehicle is in motion and $U_J$ can be used to simulate the maximum static displacement of the bridge by loading a vehicle at the mid-span of the bridge with the same dynamic load.

### 4.1. Effect of Vehicle Speed on the Impact Coefficient of a Modified Bridge

The results in Section 3.2.1 show that vehicle speed is a key influencing factor in the vehicle–bridge coupling system. In order to study the bridge after a single-pier jacking, the impact coefficients were calculated for different vehicle speeds. The impact coefficients for vehicle speeds of 30 km/h, 40 km/h, 50 km/h, 60 km/h, 70 km/h, 80 km/h, and 90 km/h are shown in Table 4:

**Table 4.** Impact coefficients at different vehicle speeds.

| Speed/(km/h) | 30 | 40 | 50 | 60 | 70 | 80 | 90 |
|---|---|---|---|---|---|---|---|
| Impact factor $\mu$ | 0.15 | 0.24 | 0.43 | 0.67 | 0.81 | 0.87 | 0.85 |
| Growth rate (%) | -- | 60% | 79% | 56% | 21% | 8% | −2.2% |

As seen from Table 4, as the speed of the vehicle increases, the impact coefficient of the bridge increases; from 30 km/h to 50 km/h, the impact coefficient increases gradually, reaching a maximum of 79%. When the speed exceeds 50 km/h, the increase in the impact coefficient starts to decrease, and even when the speed reaches 90 km/h, the impact coefficient no longer increases but instead tends to decrease. This phenomenon is because when the speed is low, the vehicle has enough time to transmit the inertial force to the bridge. However, when the speed is too high, the bridge has already left its original position before it can react. So, when the vehicle speed is too high, the impact coefficient becomes smaller.

### 4.2. Effect of Vehicle Weight on the Impact Coefficient of a Bridge before and after Modification

The impact factors for bridges at different vehicle weights are shown in Table 5:

**Table 5.** Impact coefficients before and after jacking modifications at different vehicle weights.

| Vehicle Weight | Empty Vehicles | Fully Loaded | Overloading |
|---|---|---|---|
| Impact factor before jacking $\mu$ | 0.12 | 0.65 | 0.69 |
| Impact factor after jacking $\mu$ | 0.13 | 0.67 | 0.72 |

Table 5 shows that at different vehicle weights, the impact coefficient of the bridge increases as the body weight increases. The impact coefficient of the bridge becomes larger after the single-pier jacking modification than before the jacking, but the difference is minimal. This is because before jacking the abutment, the elevation of the abutment is lower than that of the adjacent piers due to settlement, which results in the inertial forces being broken down into two parts, along the bridge in the cis direction and onto the bridge in the vertical direction, thus reducing the impact of the vehicle on the bridge when travelling. The single-pier jacking of the bridge increases the impact factor but has less impact.

*4.3. Effect of Vehicle Deflection on the Impact Coefficient of a Modified Bridge*

The exactly loaded vehicle in the off-load case results in a reduction in deflection in the center span of the bridge. However, it causes the deflection in the lane where the vehicle is located to be greater than the deflection in the span where it is loaded into the center lane. The impact coefficients of the lanes where they are located for the different deflection load cases are shown in Table 6:

**Table 6.** Bridge impact factors for different deflection loads.

| | Eccentric Position | Lane 4 | Lane 5 | Lane 6 |
|---|---|---|---|---|
| Impact factor $\mu$ | Bridge median span | 0.67 | 0.68 | 0.71 |
| | Span center of the lane | 0.69 | 0.72 | 0.76 |

As the eccentricity distance increases, the impact coefficients at the center of the bridge centerline span and the impact coefficients at the center of the lane where the vehicles are located increase, and the magnitude of the increase in impact coefficients increases with the increase in eccentricity distance. Regarding the overall change in impact coefficients at both locations, the increase in impact coefficients at the lane location where the vehicle is located is more significant than the increase in impact coefficients at the center line of the bridge. Therefore, the impact coefficient of the lane where the vehicle is located is more influenced by the vehicle travelling off-load, which is not conducive to the bridge's safety.

*4.4. Impact Coefficients for Different Bridge Deck Smoothness*

The unevenness of the bridge deck does not affect the static load results of the bridge, but the effect on the dynamic load results becomes greater. According to the definition of the impact coefficient, a poorer smoothness of the deck results in a more significant dynamic load than a smooth deck, increasing the impact coefficient of the bridge. The impact coefficients under each unevenness class of the bridge deck at a speed of 60 km/h are shown in Table 7.

**Table 7.** Impact coefficients in spans of bridges with different levels of unevenness.

| Pavement Grade | Smooth | Grade A | Grade B | Grade C |
|---|---|---|---|---|
| Impact factor $\mu$ | 0.57 | 0.67 | 0.73 | 0.86 |

As seen from Table 7, the impact coefficient of the bridge is still significantly affected by the bridge deck's unevenness, and the bridge's impact coefficient increases by 51% when it drops from a smooth surface to a Class C surface. Therefore, in the subsequent operation of the bridge, more attention should be paid to the condition of the bridge deck, and repairs should be carried out promptly when the bridge deck has been damaged.

## 5. Conclusions

The bridge single-pier jacking retrofit will make the bridge undergo large deformation, so the bridge single-pier jacking retrofit for the bridge is a severe test; this paper uses a built model to analyze the bridge power response when the vehicle is travelling at different speeds, vehicle weights, and lanes, including the case of multi-lane traffic; secondly, the

bridge power response is compared before and after the bridge jacking retrofit; lastly, the impact coefficient of the bridge is analyzed. The effect of the single-pier jacking retrofit on the impact coefficient of the bridge was analyzed. Finally, the effect of single pier jacking on the impact coefficient of the bridge was analyzed; the following conclusions are drawn:

(1) Increases in vehicle speed, vehicle weight, and deck unevenness led to an enormous vibration response and impact coefficient of the bridge, but when the vehicle speed exceeded 80 km/h, the impact coefficient of the bridge decreased rather than increased; the more significant the distance of deflection under the same vehicle speed, load and deck unevenness level, the larger were the vibration response of the bridge and the corresponding impact coefficient.

(2) After the single-pier jacking modification, the dynamic deflection of the bridge increased from 0.6105 mm before jacking to 0.62 mm after jacking, but the increase was only 1.56%. Therefore, the single-pier jacking of the bridge had no significant effect on the excitation of the bridge when the vehicle was travelling on it; this effect could be ignored.

(3) The comparison of the dynamic response and impact coefficient before and after the bridge's single-pier jacking reveals that the bridge's single-pier jacking modification increased the bridge's dynamic and impact coefficient. However, the increase was negligible at 3.07%. Therefore, the single-pier jacking modification of the bridge increased the bridge's capacity under dynamic impact and also enhanced the durability and safety of the bridge.

(4) The single-pier jacking modification of bridges can improve the bearing capacity of bridges under dynamic impacts and simultaneously improve the durability and safety of bridges. Since the four-axle engineering transporter is the vehicle that travels the most on bridges, only one vehicle was considered in this study, and the impacts of multi-vehicle combinations or fleets of vehicles travelling on bridges might be considered in the next step. For the evaluation of driving comfort before and after the jacking of a single pier, the next step may be to consider the coupling of people–vehicles–bridges and to refine the study of driving comfort.

**Author Contributions:** Conceptualization, Z.Y. and C.Y.; methodology, Z.Y.; software, Z.Y.; validation, Z.Y., C.Y. and H.L.; formal analysis, Z.Y.; investigation, Z.Y.; resources, Z.Y.; data curation, Z.Y.; writing—original draft preparation, Z.Y.; writing—review and editing, Z.Y.; visualization, Z.Y.; supervision, Z.Y.; project administration, Z.Y.; funding acquisition, C.Y. All authors have read and agreed to the published version of the manuscript.

**Funding:** This research was funded by [Natural Science Foundation of Hebei Province] grant number [E2018201106].

**Data Availability Statement:** This research relies on the single-pier jacking reconstruction project of Dongzhuang Bridge in Xushui District, Baoding City, Hebei Province. All data used to support the results of this study are included in the article.

**Conflicts of Interest:** The authors declare that there is no conflict of interest in the publication of this article.

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
