# Peer review of "Dynamic Response of a Single-Pier Jacking of a Continuous Box Girder Bridge Based on Vehicle–Bridge Coupling"

_applsci, doi:10.3390/app13169287_

Round 1
Reviewer 1 Report
In this work, the authors studied the dynamic response of a box girder bridge when a vehicle is travelling at different speed.
After reviewing this manuscript, I cannot recommend publishing this work in the journal:
1) In general, it is difficult to follow the paper since the English language needs to be considerably improved. Furthermore, the organization and structure of the manuscript needs to be improved.
2) There are a lot of missing information across the paper. For instance, table 1 includes the displacement for different locations of the bridge. However, there is not a schematic figure of the bridge, which indicates these locations or the pier numbers. Thus, it is difficult to interpret the results later.
3) There is a lack of details regarding the simulations.
In general, it is difficult to follow the paper since the English language needs to be considerably improved. The organization and structure of the manuscript also needs to be improved.
Author Response
Dear Reviewer:
Thank you for your review and comments, we have made changes according to your suggestions; Please see the attachment
Best regards

Reviewer 2 Report
This paper presents a comparative analysis of the bridge before and after the single pier jacking, and based on the vehicle-bridge coupling principle, uses the built model to analyze the dynamic response of the bridge when the vehicle is travelling at different speeds, weights, lanes and multiple lanes.
The paper is interesting, but some parts need to be revised before the paper accepting for publication:
· The Abstract is full of details that are not required in this section. Instead, no mention about the research motivations and aims is made, as well as about the usefulness/novelty of the work. Please, revise the Abstract addressing these concerns.
· The second and third paragraph of the Introduction (state of art) are very poorly written, with many typos and incomplete sentences. Please, carefully revise this part.
· It is not clear why the girder jacking may be a problem for a bridge deck under the dynamic point of view (when vehicles pass over the bridge). Some additional comments on that aspect are needed, also to consolidate the research motivations and usefulness.
· The state of art could be enlarged (only 20 references are a low number for a research article, denoting a poor state of art). The reviewer suggests one more article that could be added and that may help the Authors in finding more references:
o Gara F., Nicoletti V., Carbonari S., Ragni L., Dall’Asta A. Dynamic monitoring of bridges during static load tests: influence of the dynamics of trucks on the modal parameters of the bridge. J Civil Struct. Health Monit., 10(2), 197-217, 2020. DOI: 10.1007/s13349-019-00376-1.
· In Section 2.1, when discussing about vehicle motions (degrees of freedom), it is common to refer to the terms “pitch”, “roll” and “yaw” instead of sinking, nodding and side-rolling motion.
· In Section 2.2, there is a typo when describing the matrices (Kb has been called damping matrix instead of stiffness matrix).
· In Section 3.1 Authors use acronyms probably to describe crucial sections (Z0#, Z1#, etc). However, it may be useful to add a schematic layout of the bridge indicating where these sections are located.
· Still in Section 3.1, Authors use the term “degrees of settlement”, expressed in mm. Can they be more explicit in describing what they mean with this term?
· In Section 3.2, when discussing about unevenness (“the default level of unevenness is A”), please add a reference where this classification is stated.
· The Conclusions are very poor. A brief recap of the work must be included, as well as the motivations that led to undertake this work. Then, the main conclusions can be discussed. It is also suggestable to add some comments about the usefulness and utilities of the presented work.
In general, the English language is quite good, but the paper is sometimes written in a rather confusing way, with sentences that are not easy to follow.
In general, the English language is quite good, but the paper is sometimes written in a rather confusing way, with sentences that are not easy to follow.
Author Response

(The authors gave the same response as above.)

Round 2
Reviewer 1 Report
There are still a lot of crucial information missing (e.g. pier location, etc.) and the language and grammar need to be improved (see e.g. "Liu Jianwei, Li Dejian et al.7 The effect of transverse displacement difference on the bridge force state during bridge jacking construction was investigated").
The language and grammar still need to be improved (see e.g. "Liu Jianwei, Li Dejian et al.7 The effect of transverse displacement difference on the bridge force state during bridge jacking construction was investigated").